# Successful Rescue of a Juvenile Humpback Whale (*Megaptera novaeangliae*) Trapped Upstream of the Rance Tidal Power Station, Brittany, France

**DOI:** 10.3390/ani15233503

**Published:** 2025-12-04

**Authors:** Oihana Olhasque, Léanne Carpentier, Matthieu Duchemin, Jean-Luc Jung, Cécile Dars, Florian Boucard, Sophie Labrut, Joëlle De Weerdt

**Affiliations:** 1Association AL LARK, 50 Rue Pierre et Marie Curie, 35260 Cancale, France; leanne.carpentier@al-lark.org (L.C.); matthieu.duchemin@dmconseilgroup.fr (M.D.); 2Observatoire Pelagis, UAR 3462, La Rochelle Université-CNRS, 5 Allées de l’Océan, 17000 La Rochelle, France; cecile.dars@univ-lr.fr; 3Institut de Systématique, Évolution, Biodiversité (ISYEB), Muséum National d’Histoire Naturelle, CNRS, Sorbonne Université, EPHE-PSL, Université des Antilles, 75005 Paris, France; jean-luc.jung@mnhn.fr; 4Station Marine de Dinard du Muséum National d’Histoire Naturelle, 35800 Dinard, France; 5EDF R&D, EDF France, 22-30 Avenue de Wagram, 75008 Paris, France; florian.boucard@edf.fr; 6LABOCEA, 7 Rue du Sabot, CS 30054, Zoopole, 22440 Ploufragan, France; sophie.labrut@labocea.fr; 7Biology Department, Vrije Universiteit Brussel (VUB), Pleinlaan 2, 1050 Brussel, Belgium; 8Association ELI-S, Éducation, Liberté, Indépendance—Scientifique, 39 Allée de Verdalle, 33470 Gujan-Mestras, France; eliscientific@gmail.com

**Keywords:** humpback whale, rescue operation, out of habitat, anthropogenic environment, tidal power station, non-invasive strategy

## Abstract

In February 2023, a juvenile humpback whale (*Megaptera novaeangliae*) became trapped upstream of the Rance Tidal Power Station in Brittany, France. A multidisciplinary team, including the French National Stranding Network, implemented an innovative non-invasive rescue strategy involving water-level adjustments and artificial tidal current generation. The whale’s movements were closely monitored by land-based and boat-based observers, and its body length was estimated between 7 and 10 m. After several attempts to guide the whale out, the rescue team successfully enabled its exit from the estuary on the second day of operations. This case demonstrates the effectiveness of coordinated, adaptive interventions for mitigating anthropogenic impacts on marine mammals in modified coastal habitats.

## 1. Introduction

The humpback whale (*Megaptera novaeangliae*) is a wide-ranging marine species undertaking one of the most extensive mammalian migrations [1,2]. They migrate annually between high-latitude summer feeding grounds and low-latitude winter breeding grounds in tropical and subtropical waters [3,4,5]. Fourteen Distinct Population Segments (DPS) are currently identified based on photo-identification and genetic information [6,7,8]. The North Atlantic Ocean is home to a single North Atlantic population with regional substructure and distinct migratory connections: the West Indies DPS [6,9,10] and the Cape Verde/West Africa DPS [6,11,12]. While satellite tracking studies provide a detailed description of migratory routes from Northern European feeding grounds to the West Indies breeding grounds [13,14], connections with the Cape Verde/West Africa DPS remain unclear. The English Channel has been suggested as a potential migratory corridor for some individuals, supported by photo-identification studies, and an increased number of sightings and strandings in recent years in the eastern North Atlantic [15,16,17,18,19,20,21].

Like other baleen whales, this species is exposed to various anthropogenic threats and infrastructures, including fishing gear entanglement and vessel strikes [22]. Rescuing large cetaceans (>6 m in length) in distress is logistically complex due to their size and weight, requires specialized equipment, and carries significant risks to both the animal and the rescue team [23,24]. Disentanglement methods at sea are well documented and standardized, contributing to high success rates in animal releases [25,26]. In contrast, live strandings of large cetaceans often leads to natural death, palliative care [27] or euthanasia [28], due to the severe injuries, prolonged suffering and high likelihood of re-stranding post-rescue [23,29,30,31]. However, cases of large cetaceans trapped in anthropogenic environments are poorly documented [32,33]. Different baleen whale species have been observed trapped in areas such as harbors, marinas, dammed rivers or estuaries, including humpback whales in Nova Scotia, Canada [34], and the Sacramento River Delta, California [35], Bryde’s whales in the Manning River, New South Wales [36], and a Minke whale in the old port of Montreal, Canada [37]. Such individuals are commonly described as out of habitat [38], erratic [20] or vagrant [38]. Several factors may lead animals to leave their natural range including climate change, loss of habitat, population growth, underwater noise, disturbance, or morbidity [38]. Understanding these factors is essential, as it may allow the implementation of mitigation measures before animals enter high-risk anthropogenic areas. Nevertheless, preventive measures are not always applied or may prove insufficient, leading some animals to become trapped. In such situations, individuals may experience situations of distress critical for their survival or welfare, often involving pain, suffering, physical injury and/or an inability to escape the situation without assistance [39]. Determining whether human intervention is relevant relies on an ethical and scientific assessment that considers the difficulty of evaluating the animal’s actual level of distress, the potential benefits of intervention and the need to compare the degree of intrusiveness to the likelihood of success to avoid unnecessary stress on the animal [24,39]. Human responses can be classified according to their degree of intrusiveness [39] ranging from the establishment of safety perimeters and indirect guidance (e.g., using acoustic deterrents and surface disturbances [35,37]), to the capture and translocation of the animal [36,38], with variable operational success. In cases where welfare cannot be ensured, palliative care or euthanasia may be required [38].

Describing successful rescue attempts is crucial, as sharing effective methods can improve individual survival rates, operational procedures and outcomes. Yet, such events are rarely documented in peer-reviewed literature, despite the need for transparent and detailed reporting to develop standardized, evidence-based rescue protocols. Publishing in scientific journals also ensures long-term data accessibility, reproducibility, and comparability, supporting best practices, responder training, and management strategies for marine mammal welfare in anthropogenic environments [38,40]. No published studies reporting non-invasive methods to rescue a baleen whale currently exist. Here, we report a rare case of a juvenile humpback whale trapped upstream of the Rance Tidal Power Station (TPS) in Brittany, France, and describe a collaborative, non-invasive rescue operation based on the artificial management of water levels and tidal currents. This paper provides a detailed chronological account of the operational framework, whale movements, and outcomes, offering insight into innovative approaches for marine mammal rescue in habitats altered by human activities.

## 2. Materials and Methods

### 2.1. Study Area

The Rance estuary is located on the Brittany coast, along the English Channel, in western France (Figure 1a,b). It is a narrow, steep-sided ria with marine waters entry, spanning approximately 20 km inshore and up to 2.5 km wide at its broadest point [41,42,43,44]. The head of the estuary is located at the Chatelier Lock (Figure 1a), which defines the upstream limit of tidal propagation and restricts the southward progression of seawater [45]. The salinity along the channel shows a clear gradient from brackish to marine waters [45]. Three main zones are observed: (i) within 0–200 m upstream of the Chatelier Lock, where salinity is below 20 g/L; (ii) between 200 and 1800 m, where the channel becomes deeper and salinity rises to 25–30 g/L; and (iii) beyond 1800 m, where salinity reaches about 35 g/L. The estuary has a strong tidal range, reaching up to 13.5 m during spring tides at its mouth (Saint-Servan, Figure 1a) [44,45]. The second largest tidal power station (TPS) in the world harnesses this hyper-tidal regime (Figure 1c). As a bidirectional plant, it generates both flood and ebb, and by maximizing the head between high and low water, it imposes artificial tidal rhythms and alters hydrodynamics [46,47,48]. It is operated and managed by Électricité de France (EDF) and supplies electricity to approximately 225,000 inhabitants [42,49]. Key features of the 750 m-long TPS include a navigational lock connecting the cities of Dinard and Saint-Malo, a 390 m-long and 33 m-wide power station housing 24 turbines and six 15 m-wide, 10 m-high sluice gates and a restricted area extending both upstream and downstream of the TPS, where navigation, nautical-subaquatic activities, and swimming are prohibited [50]. The TPS basin lies within the upstream part of this restricted area. Various maritime activities occur north of the restricted area [51,52,53], including commercial and recreational fishing, recreational boating, watersports, and commercial passenger shipping.

Given their scale and bidirectional operation, tidal power stations can alter estuarine tidal regimes, with potential effects on ecological continuity between estuary and open sea and on the migration of marine megafauna [46,49,54], including marine mammals. Occasional cases of delphinids and pinnipeds passing through the sluice gates and becoming trapped upstream of the TPS have indeed already been recorded (French National Stranding Network, *unpublished data*) [55]. As the operator, EDF follows a reporting procedure for every detection of stranded, out of habitat, erratic, or vagrant marine mammals. This procedure complies with legal requirements defined by the French Ministry of the Environment, Energy, and the Sea [56], in accordance with the national transposition of the European Marine Strategy Framework Directive [57]. To reinforce this measure, a partnership has been established since 2020 between the Rance TPS and the non-profit Association AL LARK (hereafter AL LARK), which is dedicated to marine conservation through research, public awareness, and education with a focus on marine mammals. This collaboration has enabled TPS operators (TPSO) to receive training in marine mammal species identification and to implement continuous visual monitoring on both sides of the facility.

### 2.2. Rescue Coordination

The French National Stranding Network (“Réseau National Échouages”; RNE) is a citizen science network coordinated by the Observatoire Pelagis (UAR 3462, La Rochelle University-CNRS). It relies on ~400 trained local correspondents [18] who take actions as part of their professional duties or as volunteers, following standardized protocols [58].

The trapped humpback whale was first detected by TPS operators (TPSO), who promptly notified AL LARK staff acting as local correspondents of the RNE [18]. As outlined in the RNE guidelines [39], any intervention first requires confirming that the animal is unable to return to its habitat and that action would provide a net benefit without undue risk. Based on video footage and field observations from RNE correspondents, the whale’s situation was assessed as high-risk, with the TPS infrastructure preventing its ability to return to open water under usual TPS conformation. This evaluation justified proceeding with an intervention. The RNE then established a multidisciplinary rescue team (Figure 2) involving local authorities, governmental institutions, the local marine station affiliated with the French National Museum of Natural History (“Muséum national d’Histoire naturelle”; MNHN), local naturalist organizations, a veterinarian and international marine mammal experts. The operation was coordinated through a dual system: the RNE provided guidelines remotely via continuous phone communication, while AL LARK ensured direct operational coordination on-site. Communication with the public and media was jointly managed by AL LARK and RNE employees and volunteers. The strategy was proactive, involving direct engagement with journalists and bystanders on-site to provide clear, consistent information and updates throughout the rescue operation. In total, around one hundred people were mobilized over the two-day operation.

### 2.3. Rescue Strategy

Following the RNE guidelines, the intervention approach followed the precautionary principle, favoring preventive actions and minimizing negative interference with the whale. Once the need for intervention was confirmed, three levels of action were considered: (i) surveillance supported by appropriate monitoring tools to track the whale’s position, condition and signs of distress, (ii) non-intrusive measures aimed at reducing environmental disturbances around the animal and (iii) minimally or non-intrusive operations involving only the least amount of manipulation necessary, such as guidance or relocation without capture or transport. Whenever possible, the least intrusive effective option was selected. The rescue strategy was therefore designed to minimize disturbance to the humpback whale while ensuring operational efficiency and human safety [33,59,60,61]. Direct interactions with the whale were therefore avoided. Vessel approaches were limited to a minimum distance of 100 m, and each approach lasted less than 5 min before vessels returned to standby positions. Engine use and maneuvering were kept to the minimum necessary, with idling or drifting preferred whenever possible. When active deterrence was required, controlled boat maneuvers (i.e., lining up across the river, revving engines, and creating surface disturbances) were used as minimally intrusive physical guidance measures to try to prevent the whale from moving further upstream, toward the southern part of the estuary. No aerial drones were deployed as a precaution due to the whale’s compromised condition, the confined and partially flight-restricted environment (maximum authorized flight altitude of 30 m; Géoportail) [62], and the reduced visibility due to turbidity [63]. Acoustic guidance measures, such as underwater playbacks of conspecific vocalizations [64] and banging pipes, also known as Oikomi pipes [65], were considered but not implemented as the necessary equipment was not available on-site during the operational window for intervention. Moreover, these acoustic tools lack documented effectiveness for baleen whales [35,65]. The main intervention relied on adapting TPS operations. In a normal generation configuration [63], sluice gates can be opened during the flood phase, when turbines operate in inverse turbining (i.e., electricity generation during the rising tide), are turned off or occasionally in pumping mode (i.e., to raise the estuary water level) and they remain closed during direct turbining (i.e., Electricity generation during the falling tide). For the purpose of the operation, turbines and sluice gates were managed to maintain sufficient water level in the estuary to prevent stranding, yet low enough to allow gate opening and facilitate the whale’s movement toward open water. Indeed, the gates cannot be opened when the head difference between the upstream estuary and the downstream sea exceeds 4.5 m. Beyond this threshold, the head difference generates a strong flushing effect and excessive mechanical pressure, which prevent safe gate operations.

### 2.4. Whale Monitoring

A navigation ban (Arrêté No. 2023/012) was enacted by the Atlantic maritime prefect during the operation, preventing recreational and fishing vessels and allowing access only to rescue boats within the area.

Monitoring was conducted using three 5–7 m rigid-inflatable boats operated by On-Board Observers (OBO), who maintained a minimum distance of 100 m from the whale to minimize stress, in accordance with French environmental regulations (Article L. 334-1, Code de l’Environnement, France). Each boat carried at least three OBOs, each continuously observing their designated sector (port, forward or starboard), and a dedicated pilot responsible for vessel operation and safety. OBO were chosen following their expertise in marine mammal biology, behavior and monitoring, their knowledge of the area and their responsibility for maritime safety. They therefore included authorities, a RNE veterinarian, and marine scientists from the marine station and naturalist organizations. In support, approximately 50 volunteers from RNE and AL LARK land-based citizen science networks were positioned every 500 m from Saint-Servan to Plouër-sur-Rance (Figure 1) as trained Land-Based Observers (LBO). OBO and LBO communicated via instant messaging on cell phones, while communication among OBO teams was carried out through VHF radio.

Whale surfacing events (i.e., blows or behavior in which the whale breaks the water surface [66]) were visually detected by OBOs or LBOs, and whale positions were transmitted to the rescue coordination team in real time. The corresponding GPS coordinates were recorded, and daily sighting maps were produced post-rescue using QGIS v3.34.13. When successive surfacings occurred less than five minutes apart, only the GPS position of the first event was recorded to avoid pseudoreplication and over-representation of short surfacing bouts, and to maintain consistency with the five-minute temporal resolution of hydrodynamic data. A behavioral observation protocol (e.g., scan sampling) was considered to document the whale’s activity state, respiratory rate, swimming speed, and surfacing profile following standard cetacean behavioral sampling methods [67,68,69,70,71], but it could not be implemented due to limited observation conditions.

During selected surfacing events, one OBO boat was positioned 100 m parallel to the whale to visually estimate its size compared to the boat length, visually conduct veterinary assessment, and photo-capture the dorsal fin, fluke and body using a Nikon D7500 equipped with a 150–600 mm telephoto lens (Nikon Corporation, Tokyo, Japan). These photographs were used to complement body condition assessments and to perform photo-identification [72,73], enabling individual identification and potential resightings across geographic areas [12,74,75,76]. The veterinary examination was performed by the RNE veterinarian, initially from the tidal power station platform and later from the rescue boat, to comply with the minimal-disturbance strategy. No physical contact or biopsy sampling was performed to avoid additional stress to the animal. The assessment followed established diagnostic frameworks for external body and skin condition in cetaceans [77,78,79]. Body condition was evaluated visually based on the contour of the epaxial musculature, visibility of spinous processes and depression of the dorsolateral region (“sunken flanks”). Skin condition was assessed from direct observation and high-resolution photographs, focusing on lesion distribution, extent and appearance (e.g., erosions, desquamation, pigmentation changes, and algal deposition) [80].

### 2.5. TPS Data Analysis

During the rescue, TPSO accessed real-time data on turbine and gates flowrates, TPS operational modes, and water surface elevation (WSE) measurements from tidal gauges at Saint-Suliac (WSE Estuary), immediately downstream of the TPS (WSE Sea), and immediately upstream of the TPS, within the restricted area (WSE TPS) (Figure 1). These data informed adaptive management (e.g., extending gate-opening duration, aligning openings with ebb currents while maintaining sufficient depth) to support the rescue strategy (Figure 3).

Post-rescue, Électricité de France (EDF) provided flow rate, tidal power station (TPS) operational mode, and water surface elevation (WSE) data recorded at 5 min intervals. WSE values were measured every 5 min, while flow rates were computed at the same temporal resolution using a numerical modelling approach [45], which integrates turbine and sluice gate discharge equations (Figure 3). Flow rates were aggregated by gate-opening periods, and descriptive statistics (mean ± SD) were calculated to compare flow intensity and direction across periods. Flow rates are positive when the current flows in the basin-sea direction, and negative when the current flows in the sea-basin direction. WSE time-series were resampled to a uniform temporal scale using linear interpolation to capture gradual changes and preserve tidal cycle dynamics. Georeferenced whale sightings were projected along the estuary’s longitudinal axis and synchronized with both TPS operational modes and WSE data, allowing integrated spatiotemporal visualization of movements, TPS operational management, and hydrodynamic conditions observed during rescue.

## 3. Results

### 3.1. Individual Identification

The animal was identified as a humpback whale (*Megaptera novaeangliae*) based on its distinctive morphological features, including long [81,82] and completely white flippers (typical of North Atlantic individuals [6,81], and a small dorsal fin located on a dorsal hump [81]. Its estimated body length of 7–10 m indicated that it was a juvenile [3,83].

The whale’s fluke could not be photographed as no tail diving or tail slapping behaviors were observed. However, its dorsal surface and the right side of its dorsal fin were photo-captured and submitted to participatory national (Observatoire Pelagis) and international (Flukebook [84], Happywhale [85]) photo-identification databases (Figure 4). No matches were found, preventing us from confirming from which population it originates.

### 3.2. Whale Condition and Behavior

The visual veterinary assessment indicated poor to moderate body condition, characterized by a visible post-nuchal depression, sunken flanks, atrophy of the paravertebral musculature, and prominence of the spinous processes along the thoracolumbar region (RNE veterinarian, *pers. comm*.) [86]. These features were consistent with an emaciated body condition as defined by established body condition frameworks [77,79], corresponding to a score of 0–1 [78], where visibility of the dorsal vertebrae and ribcage indicates emaciation.

Skin lesions were observed predominantly along the dorsal midline and dorsal fin. They consisted of multifocal to coalescent erosive and desquamative lesions (Figure 5a), sometimes associated with brownish material adhering to the skin surface, likely composed of algae and/or microorganisms (Figure 5b) [80]. Whitish healed areas were also visible, suggesting prior frictional trauma or cicatrization (Figure 5c), not uncommon in free-ranging humpback whales. Given the absence of biopsy, photogrammetry, or close-range examination, these findings remain qualitative and the temporal stage of the lesions (recent or chronic) could not be determined. Nonetheless, the combination of emaciated body condition and altered skin appearance indicates that the whale’s overall health was likely compromised prior to and possibly aggravated [80] during entrapment.

Behavioral data could not be quantitatively recorded because the whale’s surfacings were brief, isolated, and irregular. Each surfacing generally consisted of a single blow or a brief dorsal fin appearance followed by a prolonged dive, preventing continuous visual tracking and the systematic application of the planned scan-sampling protocol. As a result, parameters such as respiratory rate, swimming speed, or surfacing profile could not be reliably determined. The data obtained therefore consist primarily of the geographic positions of each surfacing event, as isolated GPS points rather than continuous sequences, which were used to reconstruct the whale’s movements throughout the rescue operations. Nevertheless, behavioral changes were noted over time, including increased lateral movement between the riverbanks, faster swimming, and more frequent surfacing, which were interpreted as signs of stress (RNE veterinarian, *pers. comm*.) [86]. The whale did not exhibit tail diving or tail slapping behaviors, preventing any photo-documentation of the fluke. Tail diving usually occurs when a whale makes a steep, vertical dive and is uncommon in shallow or confined waters such as the Rance estuary (average WSE = 10.37 ± 1.03 m) [87]. Tail slapping is an energy-demanding behavior associated with social or aggressive signaling [88,89]. It is not depth-dependent and would not be expected in this context unless the whale experienced strong disturbance or threat.

### 3.3. Rescue Operation & Whale Movement Pattern

#### 3.3.1. First Rescue Day

On 9 February 2023, the juvenile humpback whale entered the Rance estuary, although its exact entry time remains unknown. No whale was reported by TPS operators on the previous day. The gates remained closed overnight and were reopened between 06:00 and 09:40 the morning of 9 February. At that time, tidal and operational conditions corresponded to a rising tide under sea-to-basin turbine operations. Gate water flow was negative (inland direction) with an average discharge of −5770.8 ± 1904.3 m^3^ s^−1^. The whale was first sighted by TPSO directly upstream of the TPS (Figure 6a,b; Sighting N° 0) at 10:30, and turbine operations were immediately suspended.

The rescue team was subsequently established and OBO initiated active monitoring at 11:30. The first gate-opening period (11:20–12:15; Figure 6b,c; P1) intended to guide the whale out of the estuary was unsuccessful. During this 55 min attempt, water flow was positive (seaward direction) with an average discharge of 5429 ± 1802 m^3^/s consistent with the ebb-tide pattern (WSE Sea; Figure 6c, P1), though generated artificially and with reduced intensity.

To reduce stranding risk during the ebb tide, the gates were closed to artificially maintain a higher WSE in the estuary than at sea. At 13:29, estuary WSE was 10.67 m and sea WSE was 4.84 m (Figure 6c). At the same time, the whale was sighted 12 km south of the TPS (Figure 6a,b; sighting N° 1).

From 13:36 to 14:02, the whale swam between the riverbanks at the south of the estuary (Figure 6a,b; sightings N° 2–3) and was then lost from sight. It probably headed north before being resighted at 15:13, 2 km from the TPS and at 15:56 within the TPS basin (Figure 6a,b; sighting N° 4–5). However, the gate-opening process could not be initiated before 17:30 due to the substantial difference in WSE between the TPS basin and the sea (Figure 6c).

Between 15:56 and 17:16, the whale headed south again and was resighted at Cancaval Peak (Figure 6a,b; Sightings N° 5–6). The three rescue boats were used simultaneously during a coordinated attempt to prevent its return to the southern estuary. The boats were aligned across the river and advanced synchronously to create a visual and acoustic barrier. Various maneuvers were performed, including lining up across the river, revving engines, and creating surface disturbances, to encourage the whale to change direction. However, the whale showed no reaction (neither avoidance nor change of direction) and swam beneath the vessels without surfacing, continuing to move south. Consequently, this strategy was abandoned and not used again during the rescue operation. The whale later returned toward the TPS on its own and was sighted at 17:20 within the TPS basin (Figure 6a,b; Sighting N° 7).

Nevertheless, the whale headed south again and was sighted at 18:07 near Saint-Suliac (Figure 6a,b; sighting N° 8). Both on-board and land-based monitoring stopped at 18:30 due to lack of light and safety constraints. The second gate-opening period happened after the tidal change (18:35 to 22:10; Figure 6b,c; P2) but the whale had already been lost from visual contact by that time, and this attempt also failed to guide it out of the estuary. During the night, AL LARK maintained a watch over the water body from the tidal power station operational tower, ensuring readiness to resume coordinated operations early the next morning.

#### 3.3.2. Second Rescue Day

On 10 February 2023, on-board and land-based searching efforts started at 8:00. At 09:46 the whale was first sighted by OBO near Saint-Suliac, 7 km south of the TPS (Figure 6d,e; sighting N° 1). The gates were opened at 9:00 as the tide was rising, creating a flow from the sea to the estuary (Figure 6e,f; P3) and at 10:15, after the tidal change, the gates remained open for three hours, allowing a high average flow of 3268 ± 1151 m^3^/s toward the sea (Figure 6g; P4).

Over the next 2 h, the whale headed north toward the way out while swimming between the riverbanks (Figure 6d,e; sighting N° 2–5), and reached the TPS basin at 11:57 (Figure 6d,e; sighting N° 6). At 12:05, the whale successfully swam through the open gates (Figure 6d,e).

#### 3.3.3. Back to the Open Sea

The whale was monitored by OBO until it reached Cézembre Island (Figure 6d,e; sighting N° 8), after which it was lost from sight. During the two-day operation, 6% of the sightings were recorded by TPSO, 53% by OBO, and 41% by LBO (Figure 6a,d). No humpback whale strandings were recorded by the RNE along the French coast in the following months, suggesting that the whale could have survived its incursion into and out of the Rance estuary.

### 3.4. Tidal Power Station Management During Rescue

During the rescue operation, Électricité de France (EDF) suspended normal tidal energy production to allow adaptive gate and turbine management prioritizing the whale’s safety. No formal negotiation phase was required, since animal welfare was pre-identified as the foremost priority guiding all operational decisions. This operational adjustment resulted in an estimated production loss of 1791 MWh compared to forecast values (2230 MWh expected vs. 1578 MWh produced on 9 February; 1771 MWh expected vs. 632 MWh produced on 10 February), corresponding to approximately 0.36% of the annual average output of the Rance Tidal Power Station (500 GWh). Throughout the rescue, the TPS operated in no-production mode for 16% of the time, manual mode for 72%, and power-generation mode for 12%. Power-generation mode was used to lower the WSE Estuary and only when the whale was far from the TPS to avoid any risk of passing through the turbines. Flow rate through the turbine ranged from −2259 to 4758 m^3^/s and the mean absolute value was 709 ± 1410 m^3^/s. Through intermittent turbine use, WSE Estuary was regulated and maintained within a suitable range (min = 7.31 m; max = 11.70 m; mean = 10.37 m ± 1.03) that ensured both sufficient depth for the whale’s movement and the prevention of excessive drawdown of the Rance estuary, which could have caused ecological and operational disruptions (Figure 6c,f).

## 4. Discussion

### 4.1. Rescue Coordination & Citizen Science Contribution

The success of this operation relied on the rapid response of the rescue team coordinated by the RNE. Marine scientists provided real-time interpretation of the whale’s behavior, movement and stress indicators, allowing the team to adapt interventions to minimize disturbance. The attending veterinarian contributed expertise in welfare assessment and provided immediate advice on the whale’s physical condition and safe interaction distances. The RNE brought extensive operational experience [19,20,39] in managing cetaceans in distress, ensuring the coordination of logistics, safety, and communication among all partners.

RNE and AL LARK trained volunteers played a key role in tracking the whale and maintaining effective communication with OBO to ensure continuous monitoring and adaptation of operations in real-time. Moreover, the involvement of volunteers familiar with marine mammal monitoring and RNE procedures ensured that communication with the public and media remained accurate. This approach not only maintained public support during the rescue but also reduced the risk of misinformation. Indeed, accurate and constructive communication with the public and media is essential to ensure transparency, maintain trust in management decisions, and prevent interference during operations [90,91,92]. Clear information flow also enhances public awareness and supports social acceptance of conservation issues [93]. This case also highlights the value of well-structured citizen science networks (e.g., RNE network and AL LARK’s land-based citizen science program) in supporting marine mammal rescue efforts by raising awareness, promoting responsible public behavior [38], and enhancing operational efficiency [58,94,95,96].

The long-standing partnership between AL LARK and the Rance TPS facilitated early detection of the whale, immediate prioritization of animal welfare over energy production, and coordination of industrial operations for adaptive water-level management. This collaboration also enabled transparent data sharing for post-rescue analysis. Local authorities provided crucial logistical support (e.g., authorizing temporary navigational restrictions, providing safety boats and communication channels) and regulatory supervision. This case demonstrates the value of a multidisciplinary approach [97], where expertise from multiple fields is combined to support better collective decision-making in complex rescue scenarios. Detailed reporting of similar cases would provide valuable insight to guide future interventions.

### 4.2. Implications for Whale Condition and Behavior

The whale’s presence in the Rance estuary in early February could coincide with the general migration period of North Atlantic humpback whales. During migration, juveniles often show greater variability in their movements and are generally more prone to navigational errors and strandings than adults [9,60,98]. Several European cases similarly involve young individuals entering rivers or shallow coastal systems, suggesting that inexperience or disorientation may contribute to such events [16].

Assessing baleen whale welfare is complex, as it relies on multiple indicators such as skin and body condition, behavior, and environmental factors [78,99]. Methods including visual and photo-based scoring [78,100,101,102], behavioral observations [87,103,104,105,106,107], and tools like drones and high-resolution cameras [108,109,110,111] could therefore improve reproducibility and reliability of real-time welfare assessment. However, the rarity of rescue interventions poses its own challenges to funding and maintaining the investment, maintenance and human resources costs associated with these non-invasive methods and tools.

In this case, welfare assessment and behavioral monitoring were constrained by limited observation opportunities. The whale surfaced infrequently and often at variable distances from observers, which prevented systematic behavioral sampling and reduced the number of high-quality photographs available. These constraints limited a robust, standardized body-condition evaluation and reduced the likelihood of future resightings, due to the absence of a usable fluke photograph. In addition, key behavioral parameters such as activity state, breathing patterns or surfacing frequency could not be quantified, limiting interpretation of stress, weakness or clinical condition progression over time. Although drone imaging could have mitigated some of these limitations by providing a higher vantage point, broader spatial detection, improved photographic and behavioral coverage, it was not used, as the animal’s weakened state called for a cautious approach. The whale’s brief surfacings and the high turbidity of the estuary would also have reduced image quality, since detectability depends on factors such as water clarity and dive behavior [109]. In addition, the operation took place in a confined estuarine area where a legally authorized maximum flight altitude of 30 m applies (Géoportail), which is not consistent with recommended practices from some marine mammal drone studies [62]. These studies note that flights below 30 m may cause avoidance behavior and should be limited to essential tasks, although such responses are documented mainly in odontocetes and should be applied carefully to baleen whales given their different hearing ranges [112]. These trade-offs underscore the challenges of balancing optimal data collection with animal welfare during marine mammal rescue operations. As a result, welfare evaluation relied primarily on qualitative observations and a few opportunistic photo-based assessments, making it difficult to determine whether the rescue operation or the entrapment itself affected the whale’s welfare, especially since the animal likely exhibited poor skin and body condition before entering the estuary. Poor body condition might indeed result from disease, extended migration, prey scarcity, or anthropogenic stressors [113,114,115,116,117]. Nevertheless, prolonged stress experienced during the two-day rescue may have worsened its body condition [118], while freshwater exposure may also have worsened its skin condition [35,78,80]. Future cases would benefit from continuous and more detailed monitoring throughout rescue operations to better evaluate these potential impacts. When conditions allow, the use of aerial drones could support this effort by providing consistent behavioral monitoring, improving body-condition assessment and assisting in the detection of possible injuries or entanglement.

### 4.3. TPS Management as a Rescue Strategy

Initial attempts to prevent the whale from moving further upstream using noisy vessel maneuvers and surface disturbances proved ineffective. This is consistent with established humpback whale behavior, as individuals often approach or pass beneath vessels rather than avoid them [71,119]. In marine mammal rescue operations, physical guidance techniques such as human or vessel barriers and non-entangling nets can be used for small cetaceans in narrow or shallow areas, but they were unsuitable here due to the whale’s size, its unpredictable diving behavior, and the width of the estuary, which exceeds the operational limits of such devices. Acoustic guidance tools were also considered but not implemented due to the lack of dedicated equipment on-site and the limited evidence supporting their effectiveness for baleen whales. A previous experiment, outside of a rescue operation context, showed that playback of predator sound can induce a rapid horizontal avoidance away from the speaker in humpback whales [120]. However, in a case involving a cow-calf pair of humpback whales observed in the Sacramento River Delta, California, attempts using playbacks of alarm tones, humpback and killer whale sounds, as well as Oikomi pipes, failed to produce downstream movement by the whales [35]. Successful use of such acoustic tools come mainly from killer whale (*Orcinus orca*) management contexts [33,121], an odontocete species with a different hearing range compared to humpback whales [112].

In the absence of effective physical or acoustic guidance options, given the whale’s lack of reaction to boat-based deterrence and the confined nature of the estuary, the rescue strategy shifted toward a fully non-invasive approach. As the TPS gates formed the only possible entry and exit route, adaptive management of gate openings, turbine operations and hydrodynamic manipulation became the most suitable method to facilitate the whale’s movement downstream.

Results suggest that whale movement may have been influenced by three factors: gate opening duration, whale distance from the TPS, and tidal current velocity and direction. The whale likely entered the estuary during a prolonged gate-opening period at flood tide, when sea-to-basin currents were strongest. Conversely, rescue attempts may have failed (P1, P2, P3) due to the short gate-opening duration (P1), the excessive distance between the whale and the TPS (P2, P3), and flood-tide currents generating adverse inland flow (P2, P3). Hydrodynamic studies of the Rance estuary indicate that TPS operations significantly reshape local flow dynamics [42,45]. Flood currents are strongly modulated by sluice gate operation, which can substantially accelerate flow near the eastern channel upstream of the plant, with effects that remain relatively uniform across the water column. In contrast, ebb currents are more strongly shaped by the morphology of the estuary, with marked intensification in constricted areas such as Saint-Hubert Port [42,45]. During floods, vertical flow variability is limited except at Saint-Hubert Port and Saint-Suliac, where local geometry enhances flow heterogeneity. These combined effects mean that TPS settings can alter the strength, direction and spatial extent of dominant flow pathways within the upper estuary. Although the whale might eventually have exited during a natural ebb tide, this would have required the natural ebb to coincide with TPS operating conditions allowing turbine shutdown or reverse estuary-to-sea flow combined with gate opening, conditions that do not occur systematically under normal production cycles. In contrast, the operational adjustments implemented during P4 (extended gate opening and turbine shutdown) likely reinforced and prolonged a seaward-flow corridor at a moment when the whale was close enough to enter its influence zone. Thus, it appears that the successful exit may have been facilitated by the combination of artificial hydrodynamic reinforcement and spatial proximity of the whale, rather than by tidal phase alone. This hypothesis is consistent with previous descriptions of humpback whale distribution, known to follow predictable tidal cycles in coastal habitats [122,123,124,125]. Other studies in oceanic environments suggest that this species often follows straight migratory tracks [126,127] but may align with high-intensity currents to reduce energetic costs [128]. In a hyper-tidal, confined system like the Rance estuary, artificial modulation of tidal flow therefore likely influenced both inward and outward movements, particularly considering the whale’s compromised condition. These findings underline the importance of increased vigilance during sea-to-basin turbine operations.

### 4.4. Challenges in Quantifying Whale Responses to TPS Management

In recent years in France, the RNE carried out several interventions to assist marine mammals in distress, including out of habitat cetaceans observed in anthropogenic environments such as estuaries or rivers [18,20]. This case documents the first record of a large cetacean upstream of the Rance TPS, although marine mammals are observed annually by TPSO on both sides of the facility. Occasional cases of delphinids passing through the sluice gates were indeed recorded, including bottlenose, common and Risso’s dolphins, with outcomes ranging from mortalities to successful exits (RNE, *unpublished data*) [55]. Environmental DNA (eDNA) surveys upstream of the TPS also detected the presence of harbor porpoises, harbor seals, and grey seals [129].

On a global scale, interactions between marine mammals and tidal energy devices remain poorly studied [130,131,132,133,134,135,136,137]. Currently, a dozen TPSs are installed worldwide (located in France, Canada, South Korea, the UK, China, Russia and the Faroe Islands) [138]. However, a recent assessment identified 426 candidate sites with suitable characteristics for tidal stream energy development [139], suggesting that such interactions could become more frequent in the future. To our knowledge, only one technical report describes the response to a large cetacean trapped upstream of a tidal device, documenting a humpback whale confined for at least five days upstream of the Annapolis TPS, Canada, in 2004 [34]. The facility was shut down, navigation was restricted, and the use of whale vocalizations was considered but not applied. While some actions are comparable to the present case (e.g., navigation ban and consideration of acoustic guidance methods), our intervention differed in that TPS operations were modified to manual mode rather than shutdown completely, and artificial flow conditions were created to support the rescue. This lack of comparable cases limited the development of a robust, reproducible framework to assess the juvenile humpback whale response to human-induced hydrodynamic conditions. In this case, data including WSE measured at three tidal gauge locations, TPS modes, and flow rates recorded through turbines and gates provided useful information for real-time adaptive management, but offered limited insight for post-rescue assessment of the whale’s response to the artificial manipulation of tidal currents. Further analysis will include specialized hydrodynamic modeling tools [45,140,141,142,143] to reproduce real-time current velocity measurements along the estuary, flow intensity and direction at sighting locations. These variables will be combined with behavioral observations to test potential current effects on the whale’s movements, to produce a quantitative rather than descriptive study and to ensure full computational reproducibility [40,144,145]. In the absence of these data, we cannot state with certainty that the actions taken to modify the hydrodynamic conditions necessarily influenced the whale’s behavior or directly enabled its exit. Nevertheless, considering the tidal conditions over the two days and the limited possibilities for opening the sluice gates under usual TPS production conditions, without the intervention on these indirect parameters (artificial manipulation of the flows, turbine shutdown, and gate opening), the whale would have had a very low chance of getting out, at least not within a timeframe that would not have put it in danger, given its compromised condition. Therefore, to overcome these limitations and to objectively assess such interactions, future studies should combine high-resolution current data, advanced hydrodynamic modeling, and detailed behavioral assessment to establish standardized methods for evaluating marine mammal interactions with tidal energy devices.

## 5. Conclusions

The whale’s successful exit coincided with the prolonged opening of the sluice gates, a measure intended to create more favorable conditions for the animal to move toward open water. This operational adjustment was made possible by real-time communication between TPS operators, RNE coordinators, and field observers, who synchronized an adaptive management of the tidal device with the whale’s position and behavior. The effectiveness of this action relied on several enabling factors: the early detection of the whale’s presence, continuous on-site monitoring, and the existence of long-standing partnerships between stakeholders that facilitated rapid, trust-based coordination. Limitations included the whale’s infrequent surfacing, which constrained opportunities for photo-identification, behavior observation and welfare assessments. In addition, the lack of detailed hydrodynamic data and specialized modelling tools restricted the analysis to a descriptive level, thus limiting insights into the whale’s response to the artificial manipulation of tidal currents.

Although some features were context specific, such as the Rance’s hyper-tidal regime, the TPS technical configuration and Électricité de France’s ability to adapt TPS operations, key lessons emerge:Establishing structured, multidisciplinary coordination within local emergency protocols, especially in areas where industrial activity overlaps marine mammal habitat, can improve response efficiency, communication, and risk management;Adaptive management of hydraulic infrastructure can serve as a non-invasive and effective tool to guide water flow and support animal movement in confined environments;Integration of real-time hydrodynamic and environmental data can contextualize observed animal movements, guide management decisions during interventions and improve post-event evaluation of rescue outcomes;Systematic, non-invasive monitoring should combine land- and vessel-based observations, drone surveys, photo-identification, behavioral standardized protocols and veterinary assessment;Ensuring accurate and constructive communication with the public and media can ensure transparency and prevent misinformation;Providing consistent event reporting, including spatiotemporal data, behavioral observations, hydrodynamic conditions, operational decisions and outcomes to facilitate cross-site comparison, can ensure long-term accessibility and support the integration of validated knowledge and best-practice guidelines into national and international rescue frameworks.

This case highlights both generalizable strategies and areas for improvement, offering practical guidance to enhance future marine mammals rescue protocols in anthropogenic environments.

## Figures and Tables

**Figure 1 animals-15-03503-f001:**
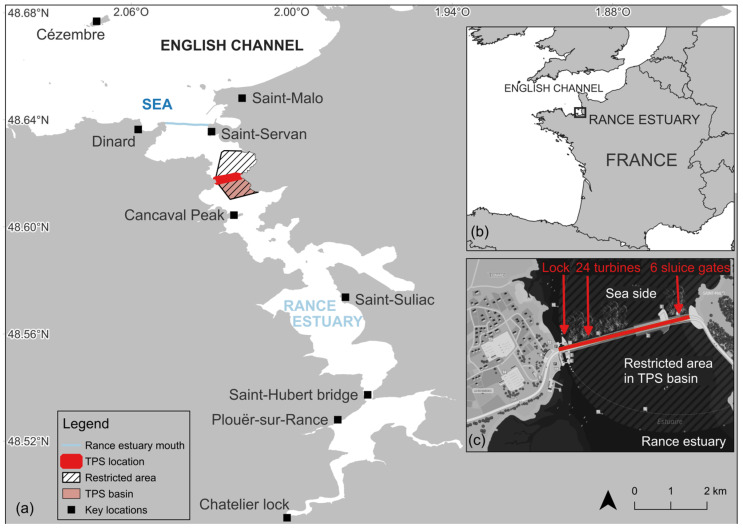
(**a**) Study area; (**b**) Location of the Rance estuary; (**c**) Tidal Power Station (TPS) © Reproduced with permission from EDF, Mémoguide; published by EDF in 2012 [50]. Data sources: GeoBretagne, Sandre, EDF France.

**Figure 2 animals-15-03503-f002:**
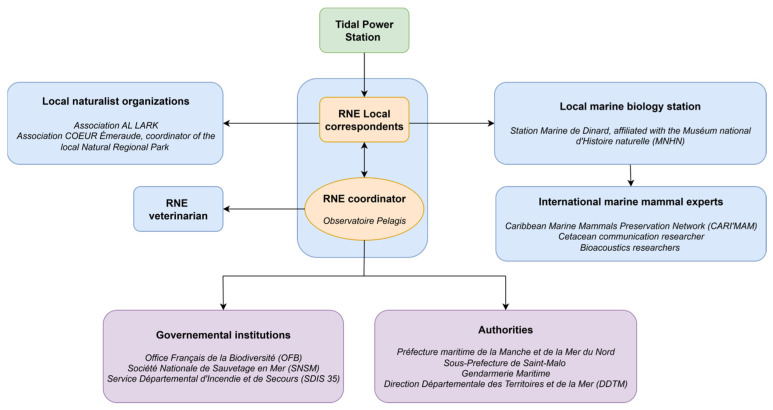
Schematic overview of the rescue team mobilization. Blue frames represent biologists, orange frames coordinators, green frame the TPS, and purple frames governmental institutions and authorities. All non-English terms refer to the official names of French institutions, organizations, or stakeholder groups and are presented in their original form to ensure accuracy.

**Figure 3 animals-15-03503-f003:**
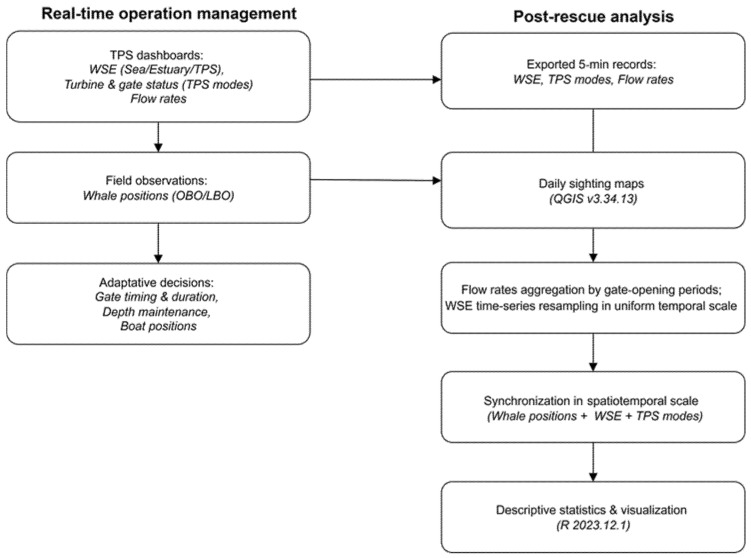
Data workflow for real-time operation management and post-rescue analysis (TPS = Tidal Power Station, WSE = Water Surface Elevation, OBO = On-board observers, LBO = Land-based observers).

**Figure 4 animals-15-03503-f004:**
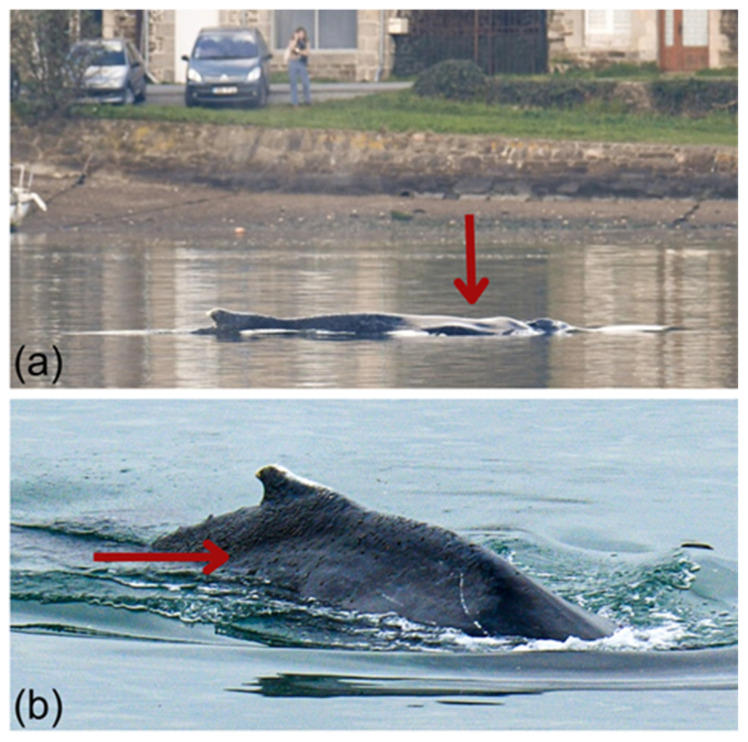
Photographs of the humpback whale trapped in the Rance estuary. Red arrows indicate external indicators of reduced body condition: (**a**) Post-nuchal depression, visible as a concave area just posterior to the blowhole, reflecting decreased blubber and epaxial muscle mass; (**b**) Dorsolateral depression (“sunken flanks”) associated with atrophy of the paravertebral musculature and increased visibility of spinous processes along the thoracolumbar region © AL LARK.

**Figure 5 animals-15-03503-f005:**
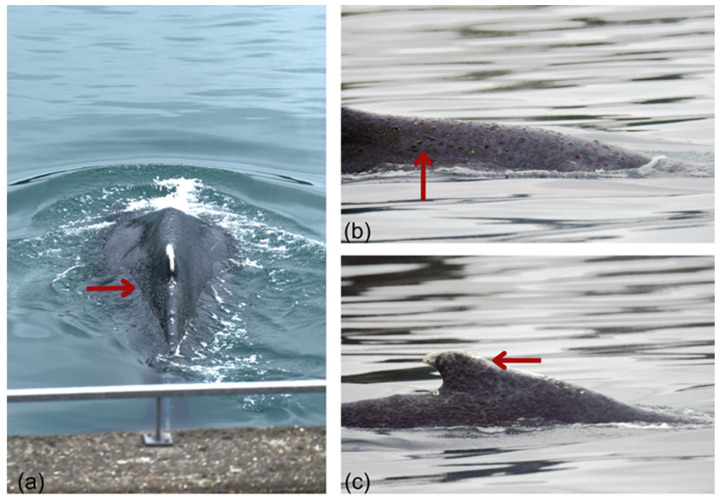
Skin condition of the humpback whale trapped in the Rance estuary. Red arrows highlight externally visible lesions distributed along the dorsal midline and dorsal fin, including: (**a**) multifocal erosive and desquamative lesions with adherent brownish material, likely algal or microbial deposits; (**b**) additional multifocal erosions and surface irregularities consistent with mild to moderate skin compromise; (**c**) irregular pigmentation and whitish areas compatible with prior frictional trauma or cicatrization. © AL LARK.

**Figure 6 animals-15-03503-f006:**
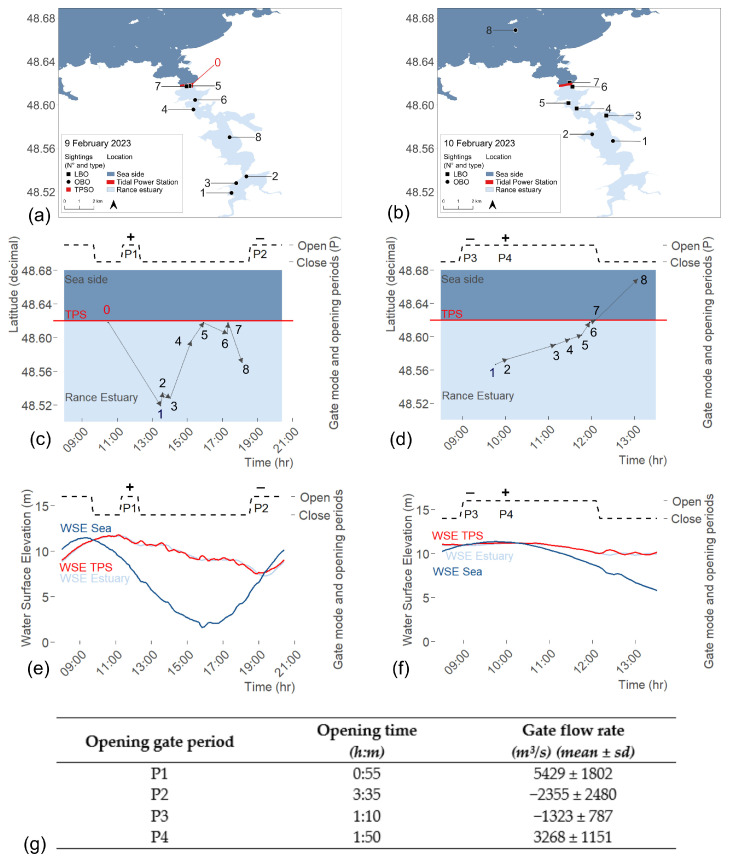
Summary of the first (**a**–**c**) and second (**d**–**f**) day of rescue; (**a**,**d**) Whale sightings and Sightings observers type (TPSO = TPS operators, OBO = On-board observers, LBO = Land-based observers); (**b**,**e**) Whale’s movements through latitudinal axis and time, and Sluice gate opening times; (**c**,**f**) Sluice gate opening times and Water Surface Elevation over time at three locations (Sea, TPS, Estuary); (**g**) Gate opening periods and measurement of flow rate through sluice gates (mean and standard deviation). Positive flows correspond to seaward direction and negative flows to landward direction.

## Data Availability

The data presented in this study are available on request from the corresponding author due to EDF data privacy restrictions.

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
