# Peer review of "Successful Rescue of a Juvenile Humpback Whale (*Megaptera novaeangliae*) Trapped Upstream of the Rance Tidal Power Station, Brittany, France"

_animals, 2025, doi:10.3390/ani15233503_

Round 1

Reviewer 1 Report

Comments and Suggestions for Authors

This article presents a successful report of a humpback whale (Megaptera novaeangliae) rescue that was stranded in a French estuary. The protocol used to lead it back to the sea involved specialists (biologists, veterinarians), volunteers from monitoring organizations, workers from the power station, and authorities who acted together following a common goal. To the knowledge of the authors, this study was the first reported case of a large mammal in the ‘Rance’ estuary and also the first non-invasive approach used to guide the whale back to the sea.

The authors and collaborators performed a wide range of operations to avoid the individual moving towards the inner part of the estuary, avoid stress and lead the way out, but they should clarify which procedures were successful and which not. They do a long description to relate all the facts and it seems a huge effort but it is unclear what would have happened if nothing at all was done. Maybe including a comparison table with reporting maneuvers could help showing relevant steps. At the same time, it seems ambitious to stablish a cause-effect relationship between the openings and flow rates of gates and the scape of the animal or the ‘three key factors’ explained in Line 345, which just one animal and 4 opening gates periods.

In the introduction, a paragraph describing main reasons of stranded or erratic animals found in estuaries is missing; so mitigation measures can be also applied prior to their occurrence.

Minor comments

Line 78: Merge the paragraph reporting interventions with the precedent describing humpback whales’ accidents.

Lines 138-9: State the roles of people involved in operations (authorities, scientists, volunteers, power station workers).

Line 209: Include species and common name of the stranded animal in results

Section 3.3: Correctly cite figure letters correspondingly: sightings should be cited in both a & b or d & e. Also check why f-g are cited before d-e. Please revise and correct.

L 315. Clarify how much energy production was lost to guarantee the survival of the cetacean and important steps in negotiations to prioritize animal welfare.

L 526 – 530: Correct reference 34.

Reviewer 2 Report

Comments and Suggestions for Authors

Abstract:

The abstract could better emphasise why this report provides scientific and practical value. While it clearly outlines the chronology and collaboration involved in the rescue, the opening might more explicitly highlight the importance of documenting such rare events in peer-reviewed literature. Since the circumstances at the Rance Tidal Power Station are highly unusual and may not apply to most other whale habitats it would strengthen the abstract to briefly state what broader insights this case offers for improving marine mammal stranding or rescue responses in anthropogenic environments, and why these lessons matter. Doing so would help readers quickly grasp the contribution of this detailed report beyond the local context. The authors talk about the value of multidisciplinary coordination, that is pretty evident and not really novel information. Therefore the bigger picture is needed with implications of how we must be capable of responding fast to the growing anthropogenic stressors in these animals' ocean environment.

Introduction:

Lines 53-54: 

The manuscript mentions two breeding populations of humpback whales in the North Atlantic. Current evidence provided in the article does not confirm this, and still, there is no published genetic evidence clearly supporting that statement. Several studies (e.g., Stevick et al. 2006, 2018; Wenzel et al. 2020) show regional structuring and different migratory connections, especially between the West Indies and Cape Verde, but there is no clear proof of reproductive isolation. More recent work (Kettemer et al. 2022) and the NOAA status review indicate that North Atlantic humpback whales are still considered one population with regional substructure. I suggest rephrasing this statement to reflect the present understanding and avoid implying that two distinct breeding populations have been established.

The introduction needs clearer arguments for the rescue attempt and the value of reporting on such a case in a peer-reviewed article. I myself understand the need, but the general reader of this journal needs to understand why it is important to improve our knowledge on effective whale stranding and rescuing procedures. The authors do include that it is crucial to share effecting protocols, but at this point it is not clear to the reader why that needs to be done via peer-reviewed research publication

Methods

Line 145:

The statement “The rescue strategy was designed to minimize disturbance” would benefit from more detail. The cited references (Similä et al.; Hunt et al. 2008) provide a general context for minimising disturbance and risk, but they do not describe the specific strategy applied in this case. It would strengthen the paper if the authors briefly outlined the measures taken to achieve minimal disturbance, e.g., limiting vessel approach, avoiding direct contact, managing noise levels, or timing operations to coincide with natural tides, so readers can understand how this principle was implemented in practice. Also, the references do not apply to humpback whales, this reference could provide more species specific focus: https://doi.org/10.2112/SI75-180.1 and the Encyclopedia of Marine Mammals provides a solid overview of stranding responses (https://doi.org/10.1016/B978-0-12-804327-1.00249-1)

Chapter 2.4

Methods of how body condition is assessed by the vets should be included

Line 177-178: Why was only the first breathing recorded in successive surfacings that occurred less than five minutes apart ?

The results chapter provides very limited data from the behavioural observations apart from geographic location. Also, the value of these observations are not clarified, only vaguely explained.

Results:

Line 209: The white pectoral fin does not reveal whether the animal is a juvenile or not, but that how the sentence may be read. You should start by explaining the characteristics of humpback whales that reveal the species identity, e.g. white pectoral fins (primarily in the North Atlantic, in other regions, dark upperside is more common), small dorsal fin located on a dorsal hump or lump, and a row of bumps (tubercles) on top of the rostrum.

Chapter 3.2

How was the emaciation assessed? The authors talk about “moderate emaciation” without referring to assessment tool which would provide this result.

Also, it would be very useful to provide a summary of the whales surfacing profile (e.g. per 10 minutes of observation), fast or slow swimming (if possible) and the distance it swam

Line 229: Tail slapping is an energy-demanding behaviour and is not restricted to deeper waters. Lifting the fluke during diving primarily happens when the whale heads vertically down to dive, which would understandably be affected by the water depth. Tail slapping is mainly associated with aggressive behaviour, or signalling to other humpbacks, therefore, it would not be expected that a whale in this situation would be tail slapping unless threatened.

Figure 4 b) the picture is too dark, the skin lesions can not be seen. The skin leasions seen in the figures (white markings) are not unusual for humpbacks.

The whale enters the river in early February which is the time of migration among humpback whales - you could elaborate more on that in the discussions. Studies have found that juvenile stranding is more likely than adult stranding I recommend using the Meyneke and Meager 2016 (https://doi.org/10.2112/SI75-180.1) to provide a better overview of what is known about stranding risks in this species elsewhere. Also, Warlick et al 2022 (https://www.frontiersin.org/journals/marine-science/articles/10.3389/fmars.2022.758812/full) provides a useful insight.

Line 245: What technique was used precisely to drive the whale out? It would be useful to report in more detail on this attempt to learn why it was not effective. It is known that humpback whales approach boats and easily surface below them, so it may not have been a surprise that the whale swam under the boat. More than one boat would have been needed since they are usually more repelled from a group of boats, according to behavioural studies on boat and humpback whale interaction

Line 297: What is a suitable range, you do provide the numbers but why is this a suitable range?

Discussions:

The discussions merely provide shallow statements about the importance of the rapid response of the team, and that the team benefited from having specialists like a veterinarian, marine scientists and the RNE. How did these specialisations provide essential input to the whale's rescue? I understand their value but the authors should not assume the general reader will understand how they contributed to the rescue and why it was valuable to obtain their assessment of the animal during the rescue.

Lines 309-313

You should provide a clearer explanation of why this is important

Line 317-318: What crucial logistical support was provided by the local authorities?

Conclusions:

The learning outcomes are not fully clear. The authors provide statements about several successful factors, but fail to clarify why and how they were successful.

Lines 389-391 summarise the main pillars that ensure the whale’s exit, according to the authors. It feels, however, as though the authors might be overselling it. It is at least not really clear to the reader what fundamentally secured the exit, apart from the fact that the tidal gate was kept open longer in the later attempt, which seemed to facilitate the exit eventually. That was presumably achieved through successful communication and collaboration between the various parties involved. I recommend providing a more straightforward or more transparent narrative which allows other rescue operators who read the paper to assess what was successful and precisely why.

Authors’ recommendations (lines 400-409): Very good, this is fundamental for this paper and could serve as a possible guidance to other rescue operators.

Line 407: However, it needs some improvement. It should include the type of behavioural monitoring you would recommend, which may have been lacking from this rescue mission (e.g., drones or other operational protocols).

Line 408: What is the value of providing accurate and constructive communication to the public and media? Please elaborate on that.

Line 409: Why is consistent event reporting important, what kind of information would you log or report?

I would like to note a potentially unnecessary self-citation in the manuscript. The reference to De Weerdt, J.; Pacheco, A.S.; Calambokidis, J.; Castaneda, M.; Cheeseman, T.; Frisch-Jordán, A.; Garita Alpízar, F.; Hayslip, C.; Martínez-Loustalot, P.; Palacios, D.M.; et al. Migratory Destinations and Spatial Structuring of Humpback Whales (Megaptera Novaeangliae) Wintering off Nicaragua. Sci. Rep. 2023, 13, 15180, 583doi:10.1038/s41598-023-41923-7. which focuses on humpback whales wintering off Nicaragua in the Pacific Ocean, appears unrelated to the present case involving a whale rescue in Brittany (North Atlantic). Unless this previous paper is specifically cited for a methodological aspect first introduced there, the citation may not be justified and could represent an irrelevant self-reference.

The citation to Weerdt, J.D. A New Record of a White Humpback Whale (Megaptera Novaeangliae) in Papeete, Tahiti. J.
Mar. Biol. Assoc. U. K. 2023, 103, e78, doi:10.1017/S002531542300067X in the sentence referring to “visual and photo-based scoring methods” appears to be inappropriate. That paper is a brief marine record describing a single leucistic humpback whale sighting in Tahiti and does not present or apply any standardised photo-based scoring or welfare assessment methodology. It therefore does not substantiate the methodological claim in that context and may represent an unnecessary self-citation.

Finally, citations 18 and 49 refer to the same paper, it seems (it is in France)

This manuscript provides valuable documentation of a rare and logistically complex rescue of a humpback whale trapped in a tidal power station basin. It contributes useful operational insights and adds to the limited peer-reviewed literature on large-whale rescues in anthropogenic environments. The collaboration between institutions and the clear timeline of events are well presented.

However, the paper currently reads more as a descriptive report than as a scientific case study. Key methodological details are missing (e.g., behavioural observations, assessment of body condition, details of the “minimal-disturbance” strategy, and the precise rescue techniques used). Some biological statements, such as the reference to “two breeding populations” of North Atlantic humpbacks, are not supported by current evidence.

The paper would benefit from clearer articulation of its broader implications, what this case teaches about improving stranding response, minimising disturbance, and managing similar future incidents. These are all correctable issues, and with additional methodological detail and stronger framing of its scientific relevance, the manuscript could make a valuable contribution to the field.

Reviewer 3 Report

Comments and Suggestions for Authors

This case report presents a well-documented and clearly structured account of a rare and logistically complex rescue of a juvenile humpback whale trapped upstream of a tidal power station. The topic is original, relevant, and of broad interest to marine conservation scientists (e.g., biologists and veterinarians), and also environmental managers dealing with human–wildlife interactions in anthropogenic habitats.

The writing is clear, methodical, and follows the Animals journal’s format and style, but I recommend some minor grammatical edits (e.g., “hydrodynamic conditions during rescue” change to “hydrodynamic conditions observed during the rescue”), since it would improve the read flow. You can see these recommendations in the pdf attached.

The manuscript’s strength lies in its integration of technical hydrodynamic data, coordinated rescue logistics, and the emphasis on non-invasive intervention. It makes a valuable contribution to the literature on marine mammal rescue protocols in modified environments.

Nevertheless, I would like the authors to reply to some questions/comments that I made in the pdf and that I am making as follow:

  • The case’s novelty — use of adaptive management of a tidal power station for whale rescue — is highly significant. Would the author's consider to better highlight how this method differs from or improves upon previous interventions (e.g., in California, Nova Scotia) in the manuscript? I feel like this topic was just mentioned very softly, and in my opinion would be worth to deep into it.

  • Consider expanding the global context in the Introduction or Discussion: How frequent are large cetacean rescues near hydropower or tidal barriers worldwide? What implications does this have for future renewable energy infrastructure? Like I said before, these questions are very important when it comes to different types of marine mammal rescue, and the more information we have, the better.

Reviewer 4 Report

Comments and Suggestions for Authors

A brief summary and general comments

The authors provide a case study on a multidisciplinary, two-day effort to rescue a juvenile humpback whale trapped upstream of a tidal power station in France. Their non-invasive rescue approach involved adjusting the water levels and generating artificial tidal currents to encourage and assist the whale’s exit from the estuary. This is an informative descriptive case study with great value to the marine mammal rescue community in that it provides an overview of the activities conducted to attempt to encourage the whale to exit the estuary in a novel situation. Rescue scenarios such as this are rare and publishing them in peer reviewed literature is critical to share and improve rescue techniques around the world.

There are a few areas where the paper could be improved. Most importantly, the case study was presented as a primary research article in format (introduction, materials and methods, results, discussion). While that works for portions of the paper, it disrupted the flow of the activities by presenting how a situation was handled prior to describing the situation. I strongly recommend that the paper be reorganized to flow according to how the events unfolded. An introduction and discussion are appropriate, but the materials and methods and results sections seemed artificial and made it more difficult for me to understand what occurred.

I am also not entirely convinced that the actions taken by the TPS influenced the whale’s successful exit from the estuary and believe the manuscript could benefit from additional data to support that conclusion as well as a slightly more comprehensive discussion that may allow for the possibility that the actions of the humans may have not changed the outcome for the whale. I also felt that the health assessment fell a bit short and provide more complete recommendations in my specific comments below. The figures are nicely presented, but could benefit from a few changes suggested below. The images, in particular, do not necessarily support the stated health assessment findings and if there are images that better relay those, I encourage inclusion thereof. References span a large time period but appear relevant.

Specific comments

  • Line 104: Recommend adding a short description of activities that occur north of the TPS – fishing, recreational vessels? This would provide better context for the situation.
  • Lines 115-116: A very brief description of the type of non-profit organization that is Association AL LARK  would be helpful here to provide context (marine mammal research, eco-tourism, naturalists, education, rescue or other?)
  • Line 120: What is the boundary of the Rance Estuary? Is it that entire system from where it meets the sea inland or is it only the portion upstream of the TPS or something different? Can that please be more clearly delineated in Figure 1a?
  • Line 158: Recommend adding a parenthetical “(farther upstream)” after “south” to better orient the readers
  • Line 160: Is prefectural the proper word here?
  • Line 165: Was there only one OBO on the vessel who was also operating it? Or were there more than one OBO on each of the three vessels? Monitoring a whale while simultaneously operating a vessel provides a different level of observational coverage than a dedicated observer with a dedicated vessel operator.
  • Line 177: Why was only the first surfacing event recorded if they occurred more frequently than 5 min? Documenting surfacings and respiratory rates can be a very informative measure of health and stress, so I would think every event should have been recorded to monitor for changes accurately.
  • Lines 178-9: What was the distance from the whale to the boat?
  • Line 192: There are a lot of acronyms in this paper, consider writing out EDF here so the reader does not have to search for it.
  • Line 192: what is meant by 5 min records – were the flow rates reported every five minutes or averaged over 5 min?
  • Line 210: replace “fins” with “flippers” Were the dorsal fin shape and presence of a dorsal hump not used to speciate the whale as well?
  • Line 218: What is meant by the dorsal vertebra? The spinous process of the vertebra (these point upwards)? In Figure 4a it looks like the arrows are pointing to the transverse processes (which extend laterally from the vertebrae)? And was just one visible or were there multiples? Was the epaxial musculature sunken? That is more what I can make out from that image, but I don’t see any readily visible vertebral processes protruding. Are there higher res or more zoomed in images that could demonstrate that better?
  • Line 222: recommend changing “dorsal line” to “dorsal midline” if that is what is intended
  • Line 225: Recommend providing more context here regarding the whale’s apparent health – poor skin condition including lesions and possible algal coating, and decreased nutritional status (especially if there is true evidence of emaciation) suggest that the whale is in compromised health. Saying it could not be definitively determined is true, but the evidence suggests a potentially compromised animal and that should be stated.
  • Line 227: Did all surfacings include a respiration? Was a respirate rate taken separate from the surfacing log? Can this please be clarified?
  • Line 228: what is tail diving? Is that meant to just mean deep diving (when they lift their tails and dive deep? Some clarification would be helpful.
  • Line 233: From the images provided I would not assess that whale to be emaciated. Thin perhaps, but can better images be provided or can the arrows in Fig 4c be better positioned to indicate what the authors are using to determine emaciated vs thin? Or perhaps those are pointing to skin lesions , which are really not visible in that image. Were ribs or scapulae visible? Was there a peanut head or post-nuchal depression? I would expect to see a prominent bilateral protrusion at the level of the transverse processes along the lumbar region if this animal were truly emaciated.
  • Lines 235+: this whole section needs to come before much of what was presented in the methods to provide the situational context first in order for readers to be able to better follow what happened.
  • Line 240: Was the tide ebbing or flowing that morning the whale presumably entered the estuary? What were the flow rates? How much observer coverage was there on the previous rising tide, meaning how certain is it that the whale was not in the estuary for longer? It would make the most sense that the whale entered the estuary on an incoming tide, but I’m not sure that corresponds to the first gate opening period or does it?
  • Line 244: was the tide ebbing or flowing during this gate opening period? What were the flow rates?
  • Line 248: Recommend adding “inland” or “south” after “12km” to provide directional context
  • Line 249: There seems to be over an hour where the whale was unaccounted form between 13:29 and 14:40 – was the whale not observed during that time?
  • Line 256: Can the authors be more specific here regarding what maneuvers were attempted? Moving back and forth in a semilunar shape? Revving engines? Fast speeds, slow speeds? And how many vessels? How were they coordinating their collective movements and activities? The more detail that can be provided the more informative this paper is to the rescue community.
  • Line 261: Can you please delineate the “TPS basin” on Figure 1a?
  • Line 265: It would have been valuable to know if the whale had any reaction to the gate opening. Were the OBOs and the LBOs discontinued due to lack of light or other logistical constraints? Were the TPSO’s still observing the whale overnight?
  • Line 270: same request regarding adding a direction “7 km south of the TPS” or similar
  • Line 285 (Figure 5): Is there a more visual way to overlay the direction of the flow on the sightings maps or perhaps on Figs b & e? Can you also include a reminder of which direction the positive and negative flows are in the Figure Legend?
  • Lines 336-7: poor body condition can also be a result of disease, recommend adding this as a differential here
  • Line 339: Salinity of the various habitats was never mentioned previously. Please add into the results section if possible to provide context for this statement.
  • Line 341: Are larger vessels or higher vantage points possible to help improve resighting ability? UAS monitoring would also be high on my list of priorities to improve capacity for in this area (both for photogrammetry as well as improving resighting ability)
  • Line 352: Can you provide some context for how much the changes to the operation of the TPS may have changed the current strength or other parameters that may have increased the likelihood of the whale’s exit. Would the whale have likely exited the estuary anyway given the ebbing tide and proximity to the TPS or do the authors feel strongly that the changes to the operation were what allowed the whale to successfully exit?
  • Lines 361-2: Can “erratic individuals” be more clearly explained here? Out of habitat? Vagrant or extra-limital?
  • Line 371: Can the authors please provide some information regarding how many of these tidal energy devices exist in the world to provide context about how frequent these interactions may be in the future.

Round 2

Reviewer 1 Report

Comments and Suggestions for Authors

The authors have answered all my comments and I do not have more suggestions. Thank you for your effort.

Reviewer 2 Report

Comments and Suggestions for Authors

Thank you for the revised version and for the detailed author response. The manuscript has improved substantially, and many of the major issues raised in the first review round have been addressed effectively. The methods section is clearer and more complete, the introduction is better justified, and several previously unclear statements were corrected.

Below, I provide a small number of remaining comments that require attention before acceptance. These relate mainly to the interpretation and use of acoustic disturbance literature, and to the statements regarding drone disturbance and acoustic tools.

Lines 199–200

The cited review (ref. 59) does not provide evidence that drones cause disturbance to baleen whales. Drone noise typically falls at high frequencies well above the functional hearing range of baleen whales, observed behavioural responses in the literature relate primarily to odontocetes (e.g., dolphins, belugas), which have much higher-frequency hearing.
I recommend rephrasing this sentence to clarify that the decision to avoid drones in this rescue was precautionary due to the whale’s compromised condition and the confined environment, rather than based on documented disturbance thresholds in humpback whales.
You may also wish to acknowledge that drones can be valuable in future rescue attempts for safely assessing body condition, behaviour, or possible entanglement.

Lines 200–201

The two references cited (Brownell et al., Ohsumi) do not mention the use of whale vocalisations to influence behaviour in baleen whales, nor do they discuss Oikomi pipes. These works focus primarily on sonar avoidance.
Oikomi pipes are documented almost exclusively for odontocetes (especially killer whales), which have a very different hearing range compared to humpback whales. For accuracy, I recommend adjusting the text and clarifying that:

  • These acoustic tools were considered but lack documented effectiveness for baleen whales,
  • Examples of their use come mainly from killer whale (odontocete) management contexts.

You may find relevant documentation here:
Northwest Area Contingency Plan: SRKW Deterrence Task Force Final Report (https://nrt.org/sites/175/files/NWAC-RRT10-SRKW%20Deterrence%20Task%20Force%20Final%20Report-with%20appendices.pdf)
• NOAA Technical Memos on acoustic deterrents (http://www.nmfs.noaa.gov/pr/publications/techmemos.htm)

This clarification will strengthen the scientific accuracy of the section.

I recommend that the authors carefully evaluate the relevance of all citations before the final submission, given the few irrelevant citations found in the first and second versions of the manuscript.

The manuscript is now close to publication. Once the small clarifications above are incorporated, the article will more accurately reflect the empirical basis for the methods discussed and avoid overstating the established use of certain acoustic tools for baleen whales.

Reviewer 4 Report

Comments and Suggestions for Authors

Thank you for submitting this revision and for your careful attention to the reviewer comments and suggestions. It is clear that the authors were thoughtful in their revision. I appreciate the improved clarity in description of the geography of the area in the methods section, including the Figure 1a updates. Thank you for also clarifying the specific intervention actions taken and more detail regarding the health assessment, both are very helpful to improve the utility of this well-documented case to help inform future interventions. The flow of information in this revision is much-improved and easy to follow, thank you for those edits.

I have a few line comments that I will outline below, but the only significant critique I have remaining is that I didn’t see the discussion about the possibility that the human actions taken may not necessarily have influenced the whale’s actions. Perhaps I missed it in the discussion, but the first sentence of the conclusion provides a much more confident picture of correlation = causation regarding why the whale went out. While I think the actions taken likely increased the likelihood of the whale’s final exit, I think qualifying this as a successful rescue takes it a step too far. I do recommend softening that statement and ensuring that the discussion does contain reference to this point. The key lessons in the conclusion also need some editing to make the structure consistent and improve clarity of each point. 

Thank you again for your attention to detail and providing this much improved revision.

Line Comments:

Line 67: not sure the parenthetical reference is necessary, seems redundant given the following text in that sentence

Line 80: disentanglement at sea is not “rehabilitation,” please delete this word

Lines 81-82: There are both chemical and physical methods of large whale euthanasia that are used around the world. If the authors are referring more specifically only to whales stranded in France or Europe and no chemical euthanasia methods are used in those locations, please clarify that here. Otherwise, please remove the word “physical” prior to “euthanasia”

Lines 84-88: Incomplete sentence now, please revise.

Line 195-6: A sentence is repeated twice

Line 238: was it considered or implemented?

Lines 306-8: Most cetacean body condition frameworks use emaciated, thin, or good condition (your ref #73 for example) rather than a term like “moderately emaciated.” As much as possible it is important to strive to standardize the terminology.  A scale of 1-4 could also be used and has been published for delphinids (Joblon et al. 2014).

Line 393: I presume the whale didn’t actually surface under the vessel? Please clarify if the whale swam under the vessel vs actually surfacing and coming in contact with the vessel

Line 607: Add reference to what country the Annapolis TPS is in (Canada)

Line 621: change “to” to “with” before “behavioral observations”

Lines 633-4: This sentence indicates that the authors believe there was a causal relationship here. Recommend rewording to construe less certainty that these actions were the cause of the whale’s exit.

Lines 648-650: This is not a complete thought/lesson, please revise

Lines 654-5: Same comment as above
